# Acute Kidney Injury in Septic Patients Treated by Selected Nephrotoxic Antibiotic Agents—Pathophysiology and Biomarkers—A Review

**DOI:** 10.3390/ijms21197115

**Published:** 2020-09-26

**Authors:** Nadezda Petejova, Arnost Martinek, Josef Zadrazil, Marcela Kanova, Viktor Klementa, Radka Sigutova, Ivana Kacirova, Vladimir Hrabovsky, Zdenek Svagera, David Stejskal

**Affiliations:** 1Department of Internal Medicine, University Hospital Ostrava, 70852 Ostrava, Czech Republic; arnost.martinek@osu.cz (A.M.); vladimir.hrabovsky@fno.cz (V.H.); 2Department of Clinical Studies Faculty of Medicine, University of Ostrava, 70300 Ostrava, Czech Republic; 3Department of Internal Medicine III—Nephrology, Rheumatology and Endocrinology, University Hospital and Faculty of Medicine and Dentistry, Palacky University Olomouc, 77900 Olomouc, Czech Republic; josef.zadrazil@fnol.cz (J.Z.); viktor.klementa@fnol.cz (V.K.); 4Department of Anesthesiology and Resuscitation, University Hospital Ostrava, 70852 Ostrava, Czech Republic; marcela.kanova@fno.cz; 5Department of Laboratory Diagnostics Institute of Clinical Biochemistry and Clinical Pharmacology, University Hospital Ostrava, 70852 Ostrava, Czech Republic; radka.sigutova@fno.cz (R.S.); ivana.kacirova@fno.cz (I.K.); zdenek.svagera@fno.cz (Z.S.); david.stejskal@fno.cz (D.S.); 6Department of Biomedical Sciences Faculty of Medicine, University of Ostrava, 70300 Ostrava, Czech Republic; 7Institute of Clinical Pharmacology Faculty of Medicine, University of Ostrava, 70300 Ostrava, Czech Republic

**Keywords:** acute kidney injury, gentamicin, sepsis, miRNA, nephrotoxicity, vancomycin

## Abstract

Acute kidney injury is a common complication in critically ill patients with sepsis and/or septic shock. Further, some essential antimicrobial treatment drugs are themselves nephrotoxic. For this reason, timely diagnosis and adequate therapeutic management are paramount. Of potential acute kidney injury (AKI) biomarkers, non-protein-coding RNAs are a subject of ongoing research. This review covers the pathophysiology of vancomycin and gentamicin nephrotoxicity in particular, septic AKI and the microRNAs involved in the pathophysiology of both syndromes. PubMED, UptoDate, MEDLINE and Cochrane databases were searched, using the terms: biomarkers, acute kidney injury, antibiotic nephrotoxicity, sepsis, miRNA and nephrotoxicity. A comprehensive review describing pathophysiology and potential biomarkers of septic and toxic acute kidney injury in septic patients was conducted. In addition, five miRNAs: *miR-15a-5p*, *miR-192-5p*, *miR-155-5p*, *miR-486-5p* and *miR-423-5p* specific to septic and toxic acute kidney injury in septic patients, treated by nephrotoxic antibiotic agents (vancomycin and gentamicin) were identified. However, while these are at the stage of clinical testing, preclinical and clinical trials are needed before they can be considered useful biomarkers or therapeutic targets of AKI in the context of antibiotic nephrotoxicity or septic injury.

## 1. Introduction

Acute kidney injury (AKI) is a common and mostly severe clinical syndrome complicating a number of critical illnesses. It has a highly negative impact on patient morbidity, mortality and clinical outcome. The diagnosis is generally based on evaluation of: (1) increase in serum creatinine and/or (2) decrease in urinary output. According to the KDIGO (Kidney Disease Improving Global Outcomes) classification of 2012, the severity of urinary output deterioration to terminal stages and presentation of an anuria and serum creatinine increase to 353.6 µmol/L is the most serious stage 3 [1]. Further, in 2017, new forms of acute renal impairment were described with AKI lasting at least 7 days after insult and acute kidney disease (AKD) lasting up to 90 days. Renal impairment and serum creatinine levels that had not returned to baseline levels by 90 days resulted in the need for renal replacement therapy (RRT) and/or progression to chronic kidney disease (CKD) [2]. Timely AKI diagnosis, especially in critically ill patients, would enable clinicians to better initiate preventive measures to avoid the need for RRT and obviate the risk of CKD. A number of promising new biomarkers may be able to predict the development or worsening of AKI in intensive care. The most highlighted of these in recent years are noncoding microRNAs in these circumstances. This review focuses on the pathophysiology and potential biomarkers in the detection of AKI after nephrotoxic drugs and/or septic insults with emphasis on specific microRNAs.

## 2. Epidemiology of Acute Kidney Injury

Acute kidney injury is a relatively frequent complication in critically ill patients in ICUs, especially in those with sepsis. The incidence of AKI in these circumstances, predominantly in situations with presentation of septic shock, may be as high as 47.5% and the overall mortality in critically ill patients with AKI may be more than 60% [3]. According to recent results of a multicenter Chinese study of patients hospitalized in ICUs, the incidence of AKI was 51%, with the majority occurring on the 4th day after admission [4]. A number of factors can contribute to AKI and progression to renal failure, including cardiovascular and hepatic disorders, malignancies, hypovolemia, intoxication, drug nephrotoxicity, anemia and surgical and vascular interventions. Further, many such patients need nephrotoxic iodine contrast drugs for CT scans and other radiological examinations. Therefore, the AKI is often a consequence of multiple factors.

## 3. Pathophysiology of Sepsis-Induced Acute Kidney Injury

Sepsis is generally characterized as a life-threatening condition induced by any type of infection (e.g., bacterial, viral, mycotic) and the dysregulated response of the host organism, with subsequent organ and tissue dysfunction or failure. The diagnosis has recently been redefined according to the SEPSIS-3 consensus (The Third International Consensus Definitions for Sepsis and Septic Shock) as an increase in the SOFA (Sequential Organ Failure Assessment) score of 2 points or more. For earlier clinical decision-making, the Quick SOFA (qSOFA) criteria can be used as evaluation of: altered mental status, a respiratory rate of 22/min or greater and a systolic blood pressure of 100 mmHg or less [5]. The pathophysiology of sepsis-induced AKI appears to be multifactorial, including, among others, deleterious inflammatory cascade [6]. Underlying explanations of septic AKI development include: (1) alteration of the renal macro- and microcirculation, with subsequent endothelial dysfunction, (2) damage of renal tubular epithelial cells, (3) a change in cellular metabolic pathways and energy consumption, (4) mitochondrial injury, (5) reactive oxygen species (ROS) production and (6) cycle cell arrest [7]. However, the exact mechanism of septic AKI is still unclear. Increase in inflammatory cytokine production and activation of leukocyte activity in the context of a dysregulated immunological and inflammatory response can lead to production of intravascular microthrombi and also reduce intrarenal blood flow and oxygen delivery [8]. Regulation of the immune and adaptive immunity response in renal tubular cells occurs due to activation of the Toll-like receptor (TLRs) family in the cell membrane. There are more than 13 members of this family and they are usually activated by endotoxins. They recognize pathogen-activated molecular patterns (PAMPs) and damage-associated pathogens (DAMPs) with the promotion of leukocyte and intrinsic renal cell activation. Renal tubular cells express TLR-1, -2, -3, -4 and -6, which can be substantially involved in the pathophysiology of tubular cell damage [9,10]. The most important receptor in septic AKI pathophysiology appears to be TLR-4, that can bind the endotoxin lipopolysaccharide (LPS), leading to activation of a number of intracellular signaling pathways via the nuclear-κB (NF-κB) transcription factor. NF-κB response to endotoxin stress leads to activation and release of the inflammatory cytokines TNFα, IL-1, IL-6 and IL-8 [11]. The activation of NF-κB depends on the phosphorylation and degradation of inhibitory κB proteins, triggered by specific kinases [9]. The basic explanation of the pathophysiological pathway in septic AKI development via activation of TLR-4 receptors in proximal tubular cells very likely lies in dysregulation of tubular integrity, with induction of tight junction disruption. This process may contribute to subsequent oliguria and decrease in renal function [12]. A recent animal study (Nakano et al., 2020), where conditional knockout of TLR-4 in proximal tubular cells reduced LPS-induced paracellular leakage of filtrate into the interstitium via TLR-4 showed that the interstitial leakage and accumulation of extracellular fluids lead to anuria and diminished the efficacy of volume resuscitation, which is frequently used in septic AKI to restore renal function [13].

## 4. Biomarkers of Sepsis-Induced Acute Kidney Injury

Many potential biomarkers have been studied in recent years in the context of sepsis and septic AKI. These can be divided into: (1) standard biomarkers, (2) additional urinary and/or serum biomarkers, (3) metabolomics, (4) other experimental proteomics and (5) microRNAs (miRNAs). Generally, AKI is diagnosed by the standard use of serum creatinine concentration and urinary output, as mentioned, with additional evaluation of serum concentration of urea. In addition, we can include Neutrophile gelatinase-associated lipocalin (NGAL), Cystatin C, Kidney Injury Molecule -1 (KIM-1), Interleukin 18 (IL-18), urinary Insulin-like growth factor-binding protein-7 (IGFBP-7), urinary tissue inhibitor of metalloproteinase 2 (TIMP-2), calprotectin, urine angiotensinogen and liver fatty acid binding protein [14]. In clinical practice, especially in patients with AKI in ICUs, it is very useful to have a biomarker capable of predicting the need for RRT initiation, renal recovery or transition to chronic nephropathy. According to a meta-analysis of 63 studies comprising 15,928 critically ill patients, the best evidence was for blood NGAL and Cystatin C followed by urinary TIMP-2 and IGFBP-7 [15]. However, decision-making in the case of RRT initiation is based on a number of clinical and laboratory findings, not only biomarkers, and none of these is specific to any particular type of AKI [16]. The major limitation of biomarkers in the AKI condition lies in comparing biomarkers to serum creatinine and diuresis, the basic diagnostic tools for AKI [17].

In recent experimental animal models of septic AKI, some potential novel metabolomic biomarkers have been identified using nuclear magnetic resonance spectroscopy on urine, renal tissue and in serum. Alterations in the concentration of several metabolites have been found e.g., lactate, *N*-acetylglutamine, alanine, pyruvate, myoinositol, glutamine, valine, glucose, ascorbic acid, aminoadipic acid, *N*-acetylaspartate and betaine and these correlate with serum creatinine and NGAL [18]. Further, many heat shock proteins (HSP) families and their bioactivity are described in various kidney diseases. In ischemic, toxic or other forms of AKI, the following have been found expressed in several renal cell types (podocytes, mesangial cells, tubular cells, fibroblasts, endothelial cells, macrophages): HSP27, HSP70, HSP60, HSP47, HSP90 and HSP32 [19]. Their main role in renal cytoprotection is still under investigation. However, many of them can block the apoptotic death pathway, oxidative stress, cell proliferation and differentiation, mediation of the inflammatory response and inhibit fibrogenesis [19]. A study of 56 critically ill patients, where 17 of them suffered from AKI, revealed that urinary HSP72 levels significantly increased in the period of three days before AKI and remained elevated during AKI diagnosis [20].

## 5. MiRNAs as Biomarkers of Septic Acute Kidney Injury

Research is currently focused on miRNAs as new potential biomarkers and/or therapeutic tools for many conditions including AKI. MiRNAs are small molecules (18–31 nucleotides) of noncoding RNAs, representing a large part of genetic information not translated from the DNA matrix into final protein production. The evidence of their abundance, developmentally regulated fashion and often subcellular localization points to their important biological role in many biochemical and pathophysiological processes and pathways on the cellular and molecular level [21]. Influence on post-transcriptional gene regulation, cell metabolism, cytokine production, cell differentiation and programmed cell death are only a small percentage of miRNAs’ effects and their target genes. In the AKI condition, some act protectively and can become potential therapeutic targets but others can increase the toxic activity and renal damage. Anti-inflammatory and/or anti-apoptotic activity in AKI has been described for the following miRNAs: *miR-10a*, *miR-21*, *miR-26a*, *miR-122*, *miR-126*, *miR-146a*, *miR-199a*, *miR-296* and *miR-494* [22]. Some miRNAs involved in the pathophysiological inflammatory process of sepsis based on endotoxin (LPS) activation of TLR4 in the signaling pathway of NF-κB activation, pro-inflammatory cytokine production (IL-6, IL-1β, TNFα) and subsequent neutrophil activation, damage of endothelial permeability and tissue injury are: *miR-146 a/b*, *miR-223*, *miR-155*, *miR-203*, *miR-15a*, *miR-16*, *miR-126*, *miR-199a* and *miR-9*. Each regulates positively or negatively a different part of the biochemical cascade to final cytokine production and tissue damage according to their target genes [23,24]. In the development of septic AKI, severe metabolic alterations of tubular epithelial cells may play a crucial role via *miR-21-3p* influence on the *AKT/CDK2-FOXO1* pathway, with induction of cycle cell arrest and apoptosis [25]. According to one human study (Ge et al., 2017), many other signaling pathways are involved in septic AKI development, including oxidative stress and mitochondrial dysfunction pathways (*HIF-1*, *PI3K-Akt*, *mTOR* and *TGFβ*). In septic, critically ill patients, significantly overexpressed *miR-4321* was observed, with the predicted oxidative-stress-associated target genes: *AKT1*, *MTOR*, *NOX5*, *IL17RA* and *IL26* [26]. The mitochondrion is assumed to be a key organelle in the development of septic acute kidney injury, and has major pathophysiological significance in ROS production and apoptosis [27]. In one hybrid human and experimental study including 50 patients with sepsis, an effect was found of *miR-106a* on caspase-3 activity, *BCl-2* expression and proinflammatory cytokine production after LPS stimulation [28]. The authors found an association between *miR-106a* and an aggravation of LPS-induced inflammation, and apoptosis in sepsis-induced AKI. A target gene for *miR-106a* was established as thrombospondin *THBS2*, which takes part in a number of processes such as regulation of cell motility, death and cytoskeleton formation [28].

## 6. Medication-Induced AKI in Septic Patients

Drug-induced nephrotoxicity varies from a relatively mild form of acute tubulointerstitial nephritis (ATIN), several types of glomerulonephritis, crystal nephropathy and osmotic nephrosis to acute tubular necrosis represented mostly by severe renal impairment with the need for RRT. The last-mentioned may be associated with development of chronic nephropathy and the need for chronic hemodialysis treatment. The incidence of medication-associated nephrotoxicity accounts for approximately 18–27% of all AKI patients in US hospitals—the main causative drugs are NSAIDs, aminoglycosides, amphotericin B and calcineurin inhibitors [29].

Potentially nephrotoxic medications, which are considered essential and commonly used in sepsis and critically ill patients, are antimicrobial agents (antibiotics, antiviral and antifungal treatment), human albumin in septic shock or proton pump inhibitors to prevent stress ulcers [30] (Table 1). Iodine contrast agents are used for radiocontrast imaging examinations in septic, critically ill patients when there is need for sepsis source finding or surgical interventions [31].

The nephrotoxicity of some antimicrobial drugs is a common problem. Of these, the most nephrotoxic are: vancomycin, aminoglycosides and polymyxins, which cause acute tubular necrosis and apoptosis depending on dose, among other factors. Many antimicrobials and other drugs frequently used in critically ill patients can also cause ATIN, accounting for 60–70% of all ATIN cases [32].

## 7. Vancomycin-Induced Nephrotoxicity Pathophysiology and Biomarkers

Vancomycin is a glycopeptidic antimicrobial agent with substantial bactericidal effect on Gram-positive bacterial infections and is frequently used in the treatment of Methicillin-resistant *Staphylococcus aureus* (MRSA). It also acts against *Streptococcus* sp., *Enterococcus* sp., *Actinomyces* sp., *Clostridium* sp. and *Eubacterium* sp. Its pharmacokinetics and pharmacodynamics are time-dependent, but according to recommendations for vancomycin treatment, the best parameter for evaluating its efficacy is the ratio of the 24 h area under the curve (AUC) to the minimum inhibitory concentration of AUC/MIC ≥400 mg.h/L with an MIC of pathogen <2 mg/L [37,38]. Vancomycin is not metabolized in the human body and is eliminated renally by glomerular filtration. Its binding to plasma proteins is less than 50%, the elimination half-life ranges from 6 to 12 h and volume of distribution is 0.4–1.0 L/kg [37]. The recently revised consensus guidelines of the American Society of Health-System Pharmacists for vancomycin therapy and monitoring for serious MRSA infections (2020) recommends, in adults and pediatric patients, a daily AUC/MIC ratio between 400 to 600 mg.h/L. A higher loading dose (20–25 mg/kg based on actual body weight) should be considered in critically ill patients treated by any type of RRT or in need of continuous vancomycin infusion. In obese patients with serious infections, the maximum loading dose is 3000 mg intravenously [39]. However, vancomycin nephrotoxicity with risk of AKI development is usually associated with higher vancomycin exposure, as measured by AUC_0–24_. Additionally, a significantly increased risk for nephrotoxicity has been observed in patients with AUC_0–24_ of 563 mg.h/L [40]. However, higher trough vancomycin serum concentrations >16.5 mg/L are also at greatest risk for new onset of AKI in critically ill patients [41]. In these cases, it is very difficult, in clinical practice, to maximize antibiotic efficacy, and, at the same time, to minimize its nephrotoxicity. The pathophysiology of vancomycin nephrotoxicity has been intensively investigated mostly in experimental research. Sakamoto et al. (2017) uncovered the possible nephrotoxic effect based on peroxidation of the mitochondrial membrane cardiolipin by vancomycin-induced production of intracellular ROS and activation of apoptosis in proximal tubular cells [42]. Apoptotic cell death induced by vancomycin may be associated with activation of specific caspases: caspase 9 and caspase 3/7 and extensive ROS production [34]. In the prospective multicenter Sapphire study (NCT01209169 ClinicalTrials.gov) including 723 critically ill adult patients, the pharmacokinetics of two urinary biomarkers TIMP-2 and IGFBP7 were evaluated in patients receiving vancomycin, piperacilin-tazobactam, or their combination. The concentration of biomarkers and the risk of death or need for dialysis treatment within 9 months, were the highest in the combination group. However, the AKI progression to an aggravated stage 2/3 was comparable in vancomycin monotherapy and in the combination treatment [43]. Selected urinary AKI biomarkers—clusterin, cystatin C, NGAL/lipocalin-2, osteopontin and KIM-1—were investigated in animals receiving vancomycin. A higher vancomycin exposure presented by an AUC_0–24_ and maximum serum concentration significantly correlated with increase in urinary AKI biomarkers but did not correlate with histopathological score [44]. Serum creatinine, urinary NGAL and KIM-1 were measured in 87 patients without chronic kidney disease (12.6% developed AKI) before and during the vancomycin therapy. According to the results, both urinary biomarkers NGAL and KIM-1 successfully discriminated patients with and without vancomycin-induced AKI earlier than serum creatinine [45]. However, in one animal study (Pais et al., 2019) KIM-1 and clusterin were more sensitive to vancomycin-induced AKI than NGAL [46].

One possible explanation for vancomycin-induced AKI and tubular cell apoptosis is based on DNA methylation by activation of Methyl-CpG binding domain protein 2 (*MBD2*). Experimental inhibition of *MBD2* can downregulate *miR-301-5p* with subsequent restoration of anti-apoptosis gene expression e.g., hepatoma-derived growth factor (*HDGF*) and microphthalmia-associated transcription factor (*MITF*) and can increase MDM-4 expression for reduction of p53 [47].

## 8. Gentamicin-Induced Nephrotoxicity Pathophysiology and Biomarkers

Gentamicin is an aminoglycoside concentration-dependent antibiotic agent with bactericidal effects against Gram-negative bacterial pathogens e.g., *Escherichia coli*, *Pseudomonas aeruginosa*, *Proteus* and *Klebsiella*. Renal elimination by glomerular filtration is the predominant type of removal of unchanged gentamicin from the human organism. Gentamicin enters to some extent the proximal tubular cells, where it is accumulated in the lysosomes with alteration of their enzymes [48]. The gentamicin volume of distribution is, approximately, an equivalent to the extracellular body weight, and it generally decreases with age. The gentamicin elimination half-life in adults with physiological renal function is very short and ranges between 2 and 3 h [49]. The incidence of acute renal failure during gentamicin treatment can account for approximately 10–20% of all cases [50]. The nephrotoxicity of gentamicin is a very complex pathophysiological process with both tubular and glomerular involvement (Figure 1). Besides the cellular uptake in proximal convoluted cells and acute tubular necrosis, it can induce mesangial cell contraction and proliferation mediated by platelet-activating factor and also an increase in free intracellular calcium concentration. Moreover, it can induce activation of the phospholipase A2 enzyme and increased production of eicosanoids in mesangial cells, production of ROS and mesangial cell apoptosis [51]. For early detection of progression from minimal to moderate kidney injury in animals treated by gentamicin, the urinary biomarkers clusterin, KIM-1, Cystatin C and NGAL were compared to serum BUN and creatinine. As predictable, all of these biomarkers showed earlier onset changes than the generally used serum BUN and creatinine in AKI diagnosis [52]. Another explanation of gentamicin nephrotoxicity is experimental evaluation of toxicity biomarkers through specific gene expression associated with apoptosis or cell necrosis. Of 10 analyzed genes associated with apoptosis, in four, *TP53*, *CASP3*, *CASP8* and *CASP9*, an increase in expression was found. In addition, the regulation of these genes produced proteins capable of cleaving specific substrates leading to cell death. Moreover, a decrease in antiapoptotic genes e.g., *BCL2L1* has been found [53]. In an experimental study carried out on drug-induced, predominantly tubular (gentamicin and cisplatine) and glomerular (puromycin and doxorubicin) kidney injury, among several commonly downregulated miRNAs, *miR-143-3p* and *miR-122-5p* were proposed as potential tubular and *miR-3473* as glomerular biomarker candidates [54].

## 9. MicroRNAs Associated with AKI Induced by Sepsis or Nephrotoxic Antibiotic Therapy

### 9.1. miR-15a-5p

The pathogenesis of sepsis AKI development is challenging, and many miRNAs likely participate in the pathophysiological and biochemical pathways involved. *MiR-15a-5p* in a regulatory axis with *XIST* (X inactive specific transcript)/*CUL3* (cullin 3 gene) in septic AKI was investigated in a combined human and animal study from China with LPS as the endotoxin. The lipopolysaccharide inhibited the growth of animal podocytes besides the upregulation of *XIST* and *CUL3* and downregulated *miR-15a-5p*. The inhibition of *XIST* and *miR-15a-5p* enhanced and preserved LPS-induced apoptosis significantly, while the *miR-15a-5p* inhibitor reversed the renal cell apoptosis. Furthermore, overexpression of the *CUL3* gene considerably reduced the LPS and *miR-l5a-5p*-induced apoptosis [57]. Another explanation of sepsis pathophysiology is based on regulation of the crucial inflammatory response of damaged organs with the participation of *miR-15a-5p*. In one animal study (Lou et al., 2020), after the LPS stimulation of macrophages there was an increased expression of *miR-15a-5p* and a release of inflammatory cytokines IL-6, IL-1ß and TNFα in comparison to a control group. Moreover, it has been demonstrated that inhibition of *miR-15a-5p* can decrease the secretion of proinflammatory cytokines by blocking its targeting gene, TNFα induced protein 3-interacting protein 2 (*TNIP2*), and the NF-κB signaling pathway [58]. *MiR-15a-5p* regulates many genes affecting angiogenesis, hematopoietic cells and carcinogenesis and has the effect of suppressing inflammation and fibrosis of peritoneal mesothelial cells induced by peritoneal dialysis [59,60,61].

### 9.2. miR-192-5p

In a human study of critically ill patients with sepsis or the nonseptic systemic inflammatory response syndrome, *miR-192-5p* was one of six of the most important circulating RNAs that differentiated sepsis from the nonseptic inflammatory response. *MiR-192-5p* negatively correlated with concentrations of pro-inflammatory cytokines (IL-6, IL-1 and IL-8) and sepsis markers (e.g., CRP). However, no correlation between the *miR-192-5p* concentration and the generally used SOFA score was found [62]. In a proceeding human study, a positive correlation was revealed between *miR-192-5p* and the redox biomarker, peroxiredoxin-1, which is released by immune cells during inflammation [63]. Urinary *miR-192-5p* was studied in animals with ischemia-reperfusion-induced AKI, where its expression in urine was significantly elevated after the ischemic intervention. The results were validated with urine samples from 71 patients who underwent cardiac surgery. The elevation of *miR-192-5p* was detected earlier than KIM-1, that was previously established as a renal injury biomarker [64]. Some other experimental studies on *miR-192-5p* and renal diseases in association with diabetes, hypertension and drug nephrotoxicity, can be added to the complex clinical and pathophysiological review. For example, circulating RNA HIPK3 (homeodomain-interacting protein kinase 3) can bind *miR-192-5p* with upregulation of transcription factor *FOXO1* (forkhead box protein O1) leading to hyperglycemia and insulin resistance [65]. In the kidney, *miRNA-192-5p* contributes to protection against hypertension through the target gene *ATP1B1* (β1 subunit of Na^+^/K^+^-ATPase), and *miR-192-5p* levels are significantly decreased in humans with hypertension or hypertensive nephrosclerosis [66]. Conversely, there is contrasting data on the kidney-protective role of *miR-192-5p* in association with vancomycin-induced AKI. The antagonism of vancomycin-induced *miR-192-5p* by the miRNA inhibitor led to a decrease of apoptosis in HK2 cells. Moreover, inhibition of p53 can attenuate apoptosis by suppressing *miR-192-5p* in vancomycin-induced AKI [67].

### 9.3. miR-155-5p

According to the literature, *miR-155* plays a critical role in various pathological and physiological processes, including immunity, inflammation, infection, cancers, hematopoietic cell differentiation, cardiovascular diseases and some genetic malformations [68].

The effects of activation and suppression of *miR-155-5p* in relation to various renal diseases and in sepsis have been investigated in a number of experimental studies [69,70]. Its role in the inflammatory process has been recently studied in an in vitro model of sepsis where inhibition of *miR-155-5p* reduced the expression of IL-6 and IL-8 as pro-inflammatory cytokines by 31% and 14%, respectively. Moreover, its inhibition can reduce the release of heat shock proteins, such as HSP10, by 69%. The latter is released from damaged cells as a stress signal [69]. The HSP10 inhibits lipopolysaccharide-induced inflammatory mediator production and NF-κB activation by inhibiting Toll-like receptor signaling in cell membranes [71]. Endogenous *miR-155* participates on regulation of inflammation and is released from dendritic cells within exosomes. It is subsequently taken up by recipient dendritic cells. Exosomal *miR-155* promoted endotoxin-induced (LPS) inflammation in one study (Alexander et al., 2015) by an increase in TNFα and subsequent increase in IL-6 serum concentration [70]. Gentamicin-induced nephrotoxicity and ischemia-reperfusion injury resulted in increased *miR-155* and *miR-18* in one rodent study [72]. With a higher dose of gentamicin, more significant injury and necrosis of renal epithelial cells were observed. However, contrary to ischemic injury, with the higher dose of gentamicin (300 mg/kg), both miRNAs decreased in the urine and increased in the renal cortex and medulla. The range of *miR-155* target genes is very high, and includes genes for the regulation of e.g., mitochondrial processes, lipid metabolism, kinase-apoptotic pathways and cell proliferation [72].

Experimental modulation of gene expression in salt-sensitive hypertensive animals showed the important role of circular RNAs in the development of hypertensive kidney injury. The authors of one study (Lu et al., 2020) characterized a circular RNA called circNr1h4 derived from the *Nr1h4* (nuclear receptor subfamily 1, group H, member 4) gene that binds to *miR-155-5p* and regulates expression of its target gene—fatty acid reductase 1 (*Far1*). The reaction between *miR-155-5p* and circNr1h4 is basically competitive, where the silencing of circNr1h4 or overexpression of *miR-155-5p* considerably decreased *Far1* levels and increased ROS production. Therefore, *miR-155-5p* may be involved in the pathology of hypertensive kidney injury [73]. The involvement of *miR-155-5p* in the pathophysiological pathway to the development of diabetic kidney disease is probably explained by the signaling axis of *p53* and *sirt1* genes with regulation of autophagic and fibrotic processes in renal tubular injury. *MiR-155-5p* may be involved in the promotion of renal fibrosis under hypoxia and also in high blood glucose concentration, and is transcriptionally regulated by p53. This allows participation in the regulation of cell growth, the cell cycle, differentiation and apoptosis [74].

### 9.4. miR-486-5p

One of the most serious causes of acute kidney injury is ischemia-reperfusion injury, often resulting in tubular cell necrosis or apoptosis. In one experimental study (Viñas et al., 2016), the effect of exosomes with *miR-486-5p* derived from endothelial colony-forming cells (ECFCs) on protection against kidney injury was investigated in mice with induced renal ischemia. Infusion of ECFC exosomes into ischemic endothelial kidney cells had a strong functional and histological protective effect, associated with increased kidney *miR-486-5p* levels, decreased phosphatase and tensin homolog (*PTEN*) and activation of the *Akt* pathway [75]. In chronic kidney disease, *miR-486-5p* inhibits the forkhead transcription factor *FOXO1* by downregulation of *PTEN* phosphatase, a negative regulator of *Akt*. *FOXO1* appears to be the predominant mediator of muscle wasting in chronic nephropathy, accelerated by stimulating the ubiquitin proteasome system through activation, e.g., E3 ligases [76]. In one human study (Regmi et al., 2019) involving patients with diabetic nephropathy, decreased serum concentrations of *miR-486-5p* were found and, this negatively correlated with albuminuria, levels of fasting blood glucose and glycated hemoglobin [77].

The association between LPS-induced inflammation and *miR-486-5p* with target *FOXO1* has been studied in vitro in nucleus pulposus cells and intervertebral disc degeneration. Experimentally, it was shown that *miR-486-5p* overexpression led to a decrease of LPS-induced production of inflammatory cytokines IL-1ß, IL-6 and TNFα and protected the nucleus pulposus cells against apoptosis [78].

### 9.5. miR-423-5p

Ischemia-reperfusion-induced AKI is one possible pathophysiological process, in which *miR-423-5p* may be substantially involved, along with other circulating miRNAs. Experimentally, it has been shown that *miR-423-5p* induces endoplasmic reticulum stress and reactive oxidative stress by inhibiting the *GSTM1* (Glutathione-S-Transferase Mu 1) gene which encodes the glutathione-S-transferase M1 enzyme in ischemia-reperfusion injury [79]. Glutathione-S-transferase is a very potent detoxification enzyme that protects the renal tubular cells against oxidative stress and ROS. The considerable involvement of *miR-423-5p* in the regulation and activation of NF-κB signaling by the *TNIP2* gene has been demonstrated in patients with lupus nephritis [80]. The *TNIP2* gene increases IKKα kinase activity and phosphorylation and induces NF-κB target genes [80]. The exact pathogenesis and factors contributing to renal cell injury here are still under investigation. *MiR-423-5p* is postulated to suppress podocyte injury in conditions of high blood glucose levels. Overexpression of *miR-423-5p* by negatively regulated Nicotinamide adenine dinucleotide phosphate oxidase-4 (*NOX4*) gene can antagonize high glucose-induced podocyte injury. Moreover, it inhibits ROS production, cell apoptosis, inflammation and subsequent damage of renal cells [81].

The schematic pathophysiology of acute kidney injury in a critically ill patient with sepsis and nephrotoxic antibiotic treatment with selected miRNAs is presented in Figure 2.

## 10. Therapeutic Approaches in Septic Patients with AKI

In the therapeutic approach to AKI in septic critically ill patients, it is essential for clinicians to decide between conservative or invasive measures. The first step is generally the treatment of sepsis by wide-spectrum antimicrobial agents, according to the epidemiologically assumed microbial [30]. Accordingly, there is a need to consider surgical or other intervention to eliminate the potential source of the sepsis. AKI severity, clinical and hemodynamic status as well as metabolic alterations are the basics for RRT or conservative therapy. Many experimental AKI biomarkers including miRNAs have an ancillary role in AKI diagnosis to assist clinicians in decision-making, in primarily preventive approaches. Usually this means dose adjustment of the nephrotoxic agents, stabilization of hemodynamics with controlled volume expansion if needed, preservation of urine output, adequate nutrition and treatment of any metabolic disorder. The crucial use of additive AKI biomarkers besides serum creatinine and urea, are the subject of ongoing research. During the first phase of septic AKI, of clinical importance is optimization of fluid management with adequate fluid resuscitation and avoidance of fluid overload. Physicians have to take into account patient’s volume status, urinary output, the type of intravenous fluids and infusion rates [82]. However, fluid overload due to loss of plasmatic proteins and increase in capillary permeability, can lead to fluid accumulation with a worsening of patient clinical status. In the renal parenchyma this increases the renal venous pressure, reduces the renal perfusion pressure and glomerular filtration rate with consequent retention of salt and water [83]. Fluid resuscitation with hemodynamic stabilization and choice of fluids play an important role in the therapeutic approach to both syndromes—sepsis and AKI. The preferred solutions are saline and balanced crystalloids, whereas hydroxyethyl starches and gelatin solutions can be associated with increased risk of AKI in septic patients. In the presence of septic shock, despite adequate volume resuscitation, there is a need for vasoactive drugs to restore renal parenchymal perfusion. In the case of sepsis, for this reason, commonly used drugs are norepinephrine, dopamine, vasopressin and phenylephrine [84].

With severe AKI, metabolic alterations and worsening hemodynamic instability, there is usually an increased need for RRT initiation in intermittent (IRRT) or continuous (CRRT) form. In physical principle and type of RRT or blood purification techniques, there are dialysis, hemofiltration, hemodiafiltration, hemoadsorption by CytoSorb (for severe sepsis) or plasmapheresis. Some RRT methods can be also combined according to clinical or laboratory findings. Close patient status monitoring and adequate supportive measures in cases with absence for urgent RRT initiation are the basic steps in conservative approaches. Thus, the timing for RRT initiation does not play a substantial role in survival in critically ill patients with AKI, especially in cases where conservative approaches can be successfully used [85]. A more comprehensive view was achieved after termination of the French AKIKI (Artificial Kidney Initiation in Kidney Injury) study (ClinicalTrials.gov NCT 01932190) performed in 620 critically ill patients with acute kidney injury. No significant difference in mortality between early and delayed strategies of RRT initiation with a decrease in need for RRT in the delayed approach was found [86]. Hemoadsorption with CytoSorb can be used predominantly in patients suffering from septic shock with careful decision-making, according to the APACHE II score. The basic principle of these blood purification devices, using more effective membranes or columns incorporated in CRRT, is removal of pro-inflammatory cytokines (e.g., IL-6, IL-8, TNFα) and endotoxins to stabilize the patient’s hemodynamics and decrease the need for vasopressor therapy [87]. Preserving adequate fluid balance, net ultrafiltration, treatment dose, nutritional support and antibiotic treatment are a vital component of the therapeutic approach in critically ill septic patients [88].

In patients with septic AKI on any type of RRT treatment, what is crucial is the antimicrobial treatment and therapeutic drug monitoring where possible. Many renally eliminated antimicrobial agents in these circumstances undergo changes in pharmacokinetics/pharmacodynamics parameters, including clearance, volume of distribution, binding to plasma proteins and elimination half-life. The dose adjustment has to be individualized according to serum concentration, to achieve the required pharmacodynamics parameters, drug efficacy and decrease the risk of toxicity [89,90]. Other preventive measures such as antioxidants in the case of antibiotic nephrotoxicity are still under investigation.

## 11. Conclusions

The early diagnosis of AKI, adequate preventive and therapeutic approaches in critically ill septic patients, are still a challenge for clinicians. Some experimental AKI biomarkers are undergoing research to help clinicians with essential, timely detection of renal injury. However, to date, there is no specific biomarker for particular toxic, septic or ischemic renal damage. None of them is able to distinguish between specific insult causing the AKI. For this reason, there is considerable current attention on miRNAs and their pathophysiological role in the human organism.

## Figures and Tables

**Figure 1 ijms-21-07115-f001:**
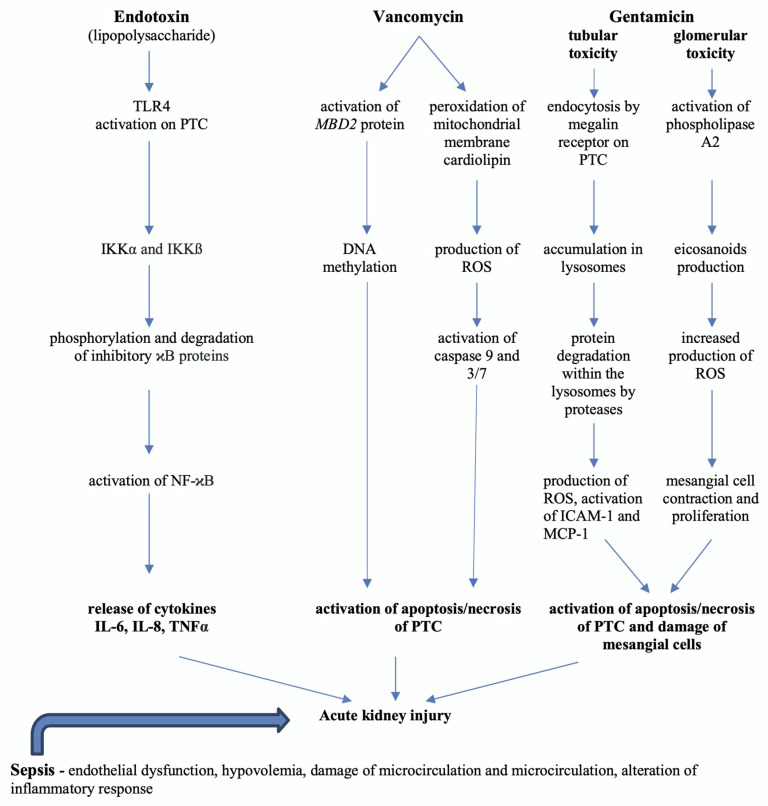
Simplified pathophysiology of acute kidney injury development in sepsis and selected antibiotic treatment [6,10,42,47,48,51,55,56]. DNA—deoxyribonucleic acid, ICAM-1—intercellular adhesion molecule-1, IKK—I-kinase, IL-6—interleukin 6, IL-8—interleukin 8, *MBD2*—Methyl-CpG Binding Domain Protein 2, MCP-1—monocyte chemoattractant protein 1, NF-κB—nuclear factor—kappa B, PTC—proximal tubular cells, ROS—reactive oxygen species, TLR4—Toll-like receptor 4, TNFα—tumor necrosis factor alpha

**Figure 2 ijms-21-07115-f002:**
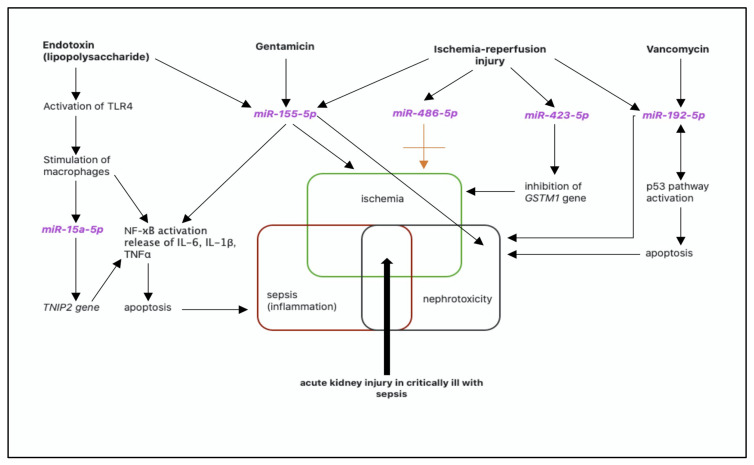
Schematic pathophysiology of acute kidney injury in a critically ill patient with sepsis and nephrotoxic antibiotic treatment with selected miRNAs. AKI—acute kidney injury, *GSTM1*—glutathione-S-transferase Mu 1 gene, IL-6—interleukin 6, IL-1ß—interleukin 1ß, NF-κB—nuclear factor—kappa B, TNFα—tumor necrosis factor alpha, *TNIP2*—tumor necrosis factor alpha induced protein 3-interacting protein 2, TLR4—Toll-like receptor 4.

**Table 1 ijms-21-07115-t001:** Nephrotoxicity of commonly used medications in critically ill patients [30,31,32,33,34,35,36].

Medications (Agents)	Clinical and Histological Presentation of Renal Toxicity
**Antimicrobials** Aminoglycosides Glycopeptides (vancomycin) Polymyxins	acute tubular necrosis, apoptosis, necroptosis, acute oxidative stress, cycle cell arrest
**Antimicrobials** β-lactams Sulphonamides Macrolides Antiretrovirals (Acyclovir) Rifampin Fluoroquinolones Chloramphenicol **Other medications** Diuretics NSAIDs Proton pump inhibitors	acute tubulointerstitial nephritis
Sulphonamides Antiretrovirals	acute crystalline nephropathy
High-osmolar iodine radiocontrast agents Hydroxyethyl starches and gelatine solutions	increase in ROS production, vasoconstriction, osmotic nephrosis
NSAIDs	altered renal hemodynamics

NSAIDs—nonsteroidal anti-inflammatory drugs, ROS—reactive oxygen species.

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
