# Peer review of "Acute Kidney Injury in Septic Patients Treated by Selected Nephrotoxic Antibiotic Agents—Pathophysiology and Biomarkers—A Review"

_ijms, 2020, doi:10.3390/ijms21197115_

Round 1

Reviewer 1 Report

The present review covers an interesting topic dealing with potential biomarkers of AKI in critically ill patients also with attention to specific micro RNAs.

While the issue is of importance and the present review is detailed, it should be restructured in some ways. It is more a review focusing on AKI pathophysiology.

-the title of the study: “Biomarkers of acute kidney injury in critically ill septic patients treated by selected nephrotoxic antibiotic agents with focus on specific microRNAs – Review” is a bit confusing or misleading:  what is meant here? The review deals more with pathophysiological aspect in general. The title of the review should perhaps be changed.

-in the review the potential nephrotoxic effects of antibiotocs are described in view of AKI-induction , on the other hand in critically ill patient their use is often necessary or mandatory. This should be commented.

- the role of microRNas as biomarker in AKI  is one subtopic in the review, but not the “focus” as stated in the title. If this is intended this should be highlighted during the review.

- in the last second paragraph titled: “ Therapeutic approaches in patients with AKI in critically ill”, here ,again the review handles with general aspects of AKI which does not parallel what is mentionned in the title of the manuscript.

Author Response

Dear reviewer,

Thank you very much for your valuable recommendations and comments.

-  manuscript has been restructured and the title has been change to AKI pathophysiology:

,,Acute kidney injury in septic patients treated by selected nephrotoxic antibiotic agents – pathophysiology and biomarkers – A review,,

- The paragraph ,, Biomarkers of sepsis induced AKI,, has been divided into two parts:

,, Biomarkers of sepsis induced acute kidney injury,, and  on page 4 has been added: ,,MiRNAs as biomarkers of septic acute kidney injury,,

- On page 5 ..,,potential nephrotoxic agents,, have been corrected - they are necessary, and reference has been added - number 30.

- the term ,,focus,, in a title of manuscript  has been deleted and title has been changed to

,,Acute kidney injury in septic patients treated by selected nephrotoxic antibiotic agents – pathophysiology and biomarkers – A review,,

 - the last second paragraph in manuscript has been corrected with focus on septic AKI - text is highlighted

With regards

Nadezda Petejova

Ostrava

Czech Republic

Reviewer 2 Report

The authors have submitted a comprehensive review of biomarkers of AKI in critically ill septic patients treated with selected nephrotoxic antibiotic agents with a focus on specific microRNAs.  The content is well organized and comprehensive. 

The manuscript requires English language review and editing and could be shortened somewhat. 

The references are appropriate and relevant.  However, I would request the authors to reference their statement on page 4 "Potentially nephrotoxic medication commonly used in sepsis and critically ill patients are antimicrobial agents (antibiotics, antiviral and antifungal treatment), intravenous immunoglobulins, human albumin, proton pump inhibitors and iodine contrast agents".  

Table 1 requires headings for each the columns.

Author Response

Dear reviewer,

Thank you very much for your valuable recommendations and comments.

- the manuscript has been English language edited - by native Canadian speaker doctor Alexander Oulton, Ph.D and by professor of English language graduated in Australia Eva Mapp, PhD. Manuscript has been shortened  on page 3 - part: ,,Pathophysiology of sepsis induced acute kidney injury,,

- the both figures have been corrected to reflect the corrections in the manuscript

- On the page 4 ,, potentially nephrotoxic medications...,, has been added reference number 30 and text has been corrected and is highlighted

- table 1 on the page 6 has been corrected with the headings added on each columns

With regards

Nadezda Petejova

Ostrava

Czech Republic

Reviewer 3 Report

First of all, the title and the abstract do not reflect the contents described in the text at all. This will cause a large confusion to the readers. I would like to suggest to rewrite all the entire parts of this manuscript by focusing the role of miRNA on AKI, mostly obtained by animal and cell culture experiments.

Author Response

Dear reviewer,

Thank you very much for your valuable recommendations and comments.

- the abstract and title of the manuscript have been changed and corrected - text is highlighted: with no focus on miRNAs, but on AKI pathophysiology and biomarkers to reflect the text of whole manuscript.

- in selected miRNAs,  these are at the stage of our clinical testing are included informations from animal and experimental study in sepsis, AKI or drug toxicity.

With regards

Nadežda Petejová

Ostrava

Czech Republic

Round 2

Reviewer 3 Report

The revised manuscript has been improved significantly, straightforward and easy to be understood.

The results described in the abstract are still needed to be improved. Comprehensive review beyond miRNA in the text is not described in the abstract at all.

Author Response

Dear reviewer,

Thank you very much for your  valuable comments and recommendations.

- The part ,, Results,, described in abstract has been corrected with highlighted AKI pathophysiology and potential biomarkers beyond miRNAs.

Kind regards

Nadezda Petejova

Ostrava

Czech Republic